



# Aerosol characteristics at the Southern Great Plains site during the HI-SCALE campaign

Jiumeng Liu[1,3], Liz Alexander[2], Jerome D. Fast[1], Rodica Lindenmaier[1], and John E. Shilling[1]

[1] Atmospheric Sciences and Global Change Division, Pacific Northwest National Laboratory, Richland, WA 99352, USA.

[2] Environmental Molecular Sciences Laboratory, Pacific Northwest National Laboratory, Richland, WA 99352, USA.

[3] School of Environment, Harbin Institute of Technology, Harbin, 150001, China.

*Correspondence to:* John E. Shilling (john.shilling@pnnl.gov)





**Abstract**

Large uncertainties exist in global climate model predictions of radiative forcing due to insufficient understanding and simplified numerical representation of cloud formation and cloud-aerosol interactions. The Holistic Interactions of Shallow Clouds, Aerosols and Land Ecosystems (HI-SCALE) campaign was conducted near the DOE's Atmospheric Radiation Measurement (ARM) Southern Great Plains (SGP) site

in north-central Oklahoma to provide a better understanding of land-atmosphere interactions, aerosol and cloud properties, and the influence of aerosol and land-atmosphere interactions on cloud formation. The HI-SCALE campaign consisted of two Intensive Observational Periods (IOPs) (April-May, and August-September, 2016), during which coincident measurements were conducted both on the G-1 aircraft platform and at the SGP ground site. In this study we focus on the observations at the SGP ground site. An

Aerodyne HR-ToF Aerosol Mass Spectrometer (AMS) and an Ionicon Proton-Transfer-Reaction Mass Spectrometer (PTR-MS) were deployed, characterizing chemistry of non-refractory aerosol and trace gases, respectively. Contributions from various aerosol sources, including biogenic and biomass burning emissions, were retrieved using factor analysis of the AMS data. In general, the organic aerosols at the SGP site was highly oxidized, with OOA identified as the dominant factor for both the spring and summer IOP

though more aged in spring. Cases of IEPOX SOA and biomass burning events were further investigated to understand additional sources of organic aerosol. Unlike other regions largely impacted by IEPOX chemistry, the IEPOX SOA at SGP was more highly oxygenated, likely due to the relatively weak local emissions of isoprene. Biogenic emissions appear to largely control the formation of OA during HI-SCALE campaign. Potential HOM (highly-oxygenated molecule) chemistry likely contributes to the highly-

oxygenated feature of aerosols at the SGP site, with impacts on new particle formation and global climate.





## 1. Introduction

Atmospheric aerosols have been the subject of intensive ongoing research due to their important impacts on the climate. They affect the climate not only through the direct scattering and absorption of solar radiation, but also by influencing the formation and properties of clouds, including radiative properties, precipitation efficiency, thickness, and lifetime (IPCC, 2013). Accurate and thorough descriptions of aerosol condensation and growth kinetics are crucial for the prediction of aerosol size distributions and therefore CCN number concentrations, which are crucial to understand for evaluating the impact of aerosols on climate (Zaveri et al., 2018;Scott et al., 2015;Riipinen et al., 2011). However, there are large uncertainties associated with cloud-aerosol interactions in global climate models (Fan et al., 2016), partly due to insufficient coincident data that couples cloud macrophysical and microphysical properties to aerosol properties. These studies demonstrate that co-located measurements of meteorology, radiation, aerosols, and clouds are needed to evaluate treatments of aerosol processes in climate models. In addition, surface processes involving land-atmosphere interactions have potential impacts on aerosol properties, which consequently influences cloud formation. To address these knowledge gaps, the Holistic Interactions of Shallow Clouds, Aerosols and Land Ecosystems (HI-SCALE) campaign was conducted near the DOE's Atmospheric Radiation Measurement (ARM) Southern Great Plains (SGP) site in north-central Oklahoma in 2016.

The Southern Great Plains (SGP) site is one of the world's largest and most extensive climate research facilities. While the SGP site is located in a rural environment with the nearest population centers approximately 40 km away (Figure S1), it is impacted by a mixture of anthropogenic, biogenic, and biomass burning sources of aerosols and their precursors. Indeed, a previous study has shown that aerosols arriving in the SGP site are of diverse origins (Parworth et al., 2015). The HI-SCALE campaign consisted of two 4-week Intensive Observational Periods (IOPs), one occurring from April 24 to May 21 (denoted as "spring" IOP) and one running from August 28 to September 24 (denoted as "summer" IOP) to take advantage of different stages and distribution of 'greenness' of cultivated crops, pasture, herbaceous, and forest vegetation types (Fast et al., 2019). One goal of the HI-SCALE campaign was to provide a detailed set of aircraft and surface measurements needed to obtain a more complete understanding and improved parameterizations of the lifecycle of aerosols and their impact on shallow clouds. To achieve these objectives, coincident measurements of meteorological, cloud, and aerosol properties collected routinely at ARM's SGP facility were augmented by additional instrumentation, in particularly by gas and particle-phase mass spectrometers.

In this study we focus on Aerosol Mass Spectrometer particle-phase measurements collected at the SGP site, with measurements of volatile organic compounds (VOC) providing insights into the precursors and help identify the sources of air parcels. The characteristics of the aerosol properties are summarized and a comparison between the spring- and summer- IOPs is shown. Potential sources of aerosols and aerosol precursors are investigated using Positive Matrix Factorization (PMF) and back trajectory analyses. In



addition, several case studies are discussed in detail to examine the impacts of seasonal variations in biogenic and anthropogenic sources, long-range transport and meteorology on aerosol properties.

65



## 2. Experimental Methods

### 2.1. SGP site description

The central facility at the SGP site is located in north-central Oklahoma, at 36.60°N and 97.48°W, as shown in Figure S1. It was designed to measure cloud, radiation, and aerosol properties in a region that experiences a wide variety of meteorological conditions. As the first field measurement site established by the Atmospheric Radiation Measurement (ARM) user facility, the SGP site is known as a "hotspot" of land-atmosphere interactions that influences the lifecycle of shallow convection (e.g., Dirmeyer et al., 2006;Koster et al., 2004;Koster et al., 2006). The central facility is located in a rural environment, immediately surrounded by cropland and pasture with a small portion of forest also in close proximity to the facility (Sisterson et al., 2016). Several urban areas are located within 200 kilometers of the site, including Wichita (~110 km to the north), Oklahoma City (~135 km to the south) and Tulsa (~150 km to the southeast). Several smaller towns such as Enid, Stillwater and Ponca City are located within 100 km of the site. In addition, a refinery is located approximately 45 km ENE of the site and a 1138 MW coal-fired power plant is located 50 km to the ESE. Therefore, the air masses arriving at the SGP site are diverse, originating from anthropogenic, biogenic, and biomass burning sources. During the two IOPs of the HI-SCALE campaign, a suite of supplemental online instruments were deployed at the SGP central facility to characterize both the gas- and particle-phase composition. Most instruments were located in the guest user facility, which is a separate trailer 300 meters from the main building of the central facility (Sisterson et al, 2016). Due to the proximity of the guest instrument trailer to the permanent aerosol equipment located at the central facility, the instruments are expected to sample the same air mass.

### 2.2 Instrumentation

An Aerodyne High-Resolution Time-of-Flight Aerosol Mass Spectrometer (abbreviated as AMS hereafter) was deployed at the SGP site to provide the mass concentration and chemical composition of submicron, non-refractory aerosols (Jayne et al., 2000;DeCarlo et al., 2006). The AMS was operated in the standard "V" mass spectrometer (MS) mode, with a 5-min data averaging interval. Filter blanks were performed every day by diverting AMS-sampled air through a HEPA filter, and these filter periods were used to account for gas-phase interferences with isobaric particulate signals. Based on the standard deviation of these blank measurements ($3\sigma$) as described in the literature (DeCarlo et al., 2006), the detection limits of the AMS at the 5-min sampling interval were 0.07, 0.015, 0.006, 0.009, and 0.005 $\mu g/m^3$ for organics, sulfate, nitrate, ammonium, and chloride, respectively. The AMS was operated continuously during the entire spring IOP and first half of the summer IOP (August 28-September 9, 2016); the AMS suffered an ion optics failure during the second half of the summer IOP. During the campaign, the AMS was routinely calibrated using monodisperse $NH_4NO_3$ particles quantified with a TSI condensation particle counter (CPC), whereas $(NH_4)_2SO_4$ particles were applied for AMS calibration before and after campaign. Data were analyzed in Igor Pro (v6.37) using the high-resolution analysis package (Squirrel v1.57, PIKA v1.16) and



techniques described in the literature (Canagaratna et al., 2015;Kroll et al., 2011;Aiken et al., 2008;Allan et al., 2004;Jimenez et al., 2003). The values of atomic oxygen-to-carbon (O:C) and hydrogen-to-carbon (H:C) ratios were calculated using the updated fragmentation tables in Canagaratna et al. (2015). Positive matrix factorization (PMF) analysis was performed using the high-resolution data and the PMF Evaluation Tool

(v3.05A).

The AMS sampled air drawn from an inlet located 10-m above the ground. Sample air was drawn through a PM$_{2.5}$ cyclone, passed through a Nafion dryer, brought into the guest facility with ½" stainless steel tubing, and shared by three aerosol sampling instruments including an HR-ToF-AMS, a Single Particle Laser Ablation Time-of-Flight mass spectrometer (SPLAT II), and an SMPS. The SMPS system consisted of a

TSI Model 3081 long column DMA with a recirculating sheath flow of 3 Lpm and a TSI Model 3775 CPC operated in the low flow mode (0.3 Lpm), and was set to measure the particle size distribution from14 nm to 710 nm (mobility diameter) at a sampling frequency of one scan every 4 minutes. Data from the SMPS were also used in evaluating the AMS collection efficiency.

An Ionicon quadrupole high-sensitivity Proton-Transfer-Reaction Mass Spectrometer (PTR-MS) was used

to measure the mixing ratios of gas-phase VOCs (Lindinger and Jordan, 1998). Similar to the aerosol sampling instrument, the inlet of PTR-MS was also positioned at 10-m above the ground with an inlet filter at the end to remove particles, then connected to the instrument through Teflon tubing. The PTR-MS was run in the mass-scan mode, in which a mass spectrum from m/z 21 to m/z 250 was recorded with 1-s dwell time on each unit m/z. The time resolution for each cycle is ~ 4-min. Drift tube pressure and temperature

were set at 2.2 mbar and 60 °C with a 600 V potential across the drift tube. Signal intensity of selected species, including m/z 42, 45, 59, 69, 71, 79, 93, 107, 121 and 137, was then converted to ppbv using a multi-point calibration with air from a calibration cylinder (Apel Riemer Environmental Inc.) containing known concentrations of acetonitrile, acetaldehyde, acetone, isoprene, methacrolein, benzene, toluene, m-xylene, trimethylbenzene (TMB), and alpha-pinene. It is assumed in our analysis that the signals at the

aforementioned m/z values are entirely from the indicated species, which could be a source of uncertainty. The calibration was performed periodically before, during, and after the campaign. The PTR-MS background was assessed twice per day by diverting air through a stainless-steel tube filled with a Shimadzu platinum catalyst heated to 600 °C, which removes VOCs from the airstream without perturbing RH. The catalyst efficiency was tested by comparing signal from air containing VOCs passed through the

catalyst with signal from VOC-free air.

### 2.3. Back trajectory analyses

To investigate the potential sources and transport pathways of aerosols and aerosol precursors observed at the SGP site, back trajectories were performed by utilizing the National Oceanic and Atmospheric Administration (NOAA) Air Resources Laboratory Hybrid Single Particle Lagrangian Integrated Trajectory

(HYSPLIT; http://ready.arl.noaa.gov/HYSPLIT_traj.php) (Draxler and Rolph, 2012). 72-hour backward



trajectories initialized from the SGP site were computed every 3 hours at multiple altitudes. The back trajectory analyses help identify sources of aerosols for specific events during the campaign.

## 3. Results and Discussion

### 3.1 Analysis of HYSPLIT trajectories

Figure 1 shows the HYSPLIT back trajectory paths for both spring- and summer- IOPs. In the spring campaign, the back trajectories suggested that the air masses arriving at the SGP site mainly originated from the north during first half of the IOP, and gradually transitioned to originating from the south. At the end of the spring IOP there was a several-day period when the dominant winds became easterly, which would bring air masses from the biogenic-rich eastern region (Parworth et al., 2015). Approximately 45%

of the time during the spring IOP, air arriving at the SGP facility originated from the northern plains. This set of trajectories passed primarily over grassland and cropland (Trishchenko et al., 2004), and would therefore be influenced primarily by weak biogenic emissions. Air masses from the south are also a major source impacting the SGP site, contributing 36% of the back trajectories. These trajectories passed by cities such as Houston and Oklahoma city, which are largely influenced by anthropogenic emissions. For the

remaining ~20% of the trajectories, air masses traveled from the east and likely brought emissions from deciduous and mixed forests in northern Arkansas, Missouri and southern Illinois. It is possible that air masses originating from the southeastern U.S. can be transported to the SGP site, but the transport period would be longer than three days.

During the summer IOP, air masses arriving at SGP site originated from two main directions (Figure 1)

according to HYSPLIT analysis. Southerly winds dominate during the summer, accounting for ~63% of the trajectories, which suggests a larger contribution of aerosols and their precursors transported from Oklahoma and eastern Texas with urban characteristics. Compared to spring IOP, a smaller fraction of the air masses originated from the north, only accounting for ~37% of the trajectories. The back trajectories during summer IOP indicate shorter transport distances than spring due to lower wind speeds.

### 3.2 Overview of the temporal variations of submicron aerosol composition and trace gases during spring- and summer- IOPs

The AMS results for non-refractory submicron aerosols (NR-PM1) observed during both the spring- and summer- IOPs are summarized in Figure 2 and Table 1. Organic aerosol (OA) contributed the largest fraction to the total NR-PM1 mass concentration during both the spring- and summer- IOPs, accounting for >60% on average. There are, however, periods where inorganics were greater than 50% of the total mass.

Average OA loading was 2.5 μg/m$^3$ in the spring and 3.8 μg/m$^3$ in the summer. Similar to organics, sulfate was also more abundant in absolute mass in the summer than in the spring IOP (average concentration 0.79 μg/m$^3$ in spring versus 1.29 μg/m$^3$ in summer; details in Table 1), but the mass fraction is similar (20.1% during spring IOP versus 22.4% during summer IOP). In contrast, the level of nitrate is much lower in the

summer IOP (0.085 µg/m$^3$) than in the spring IOP (0.244 µg/m$^3$; details in Table 1). This may be due to its semi-volatile nature with warmer temperatures pushing the equilibrium back to the gas phase, decreasing nitrate concentrations. Due to incomplete datasets of gas-phase $NH_3$, $HNO_3$ and $SO_2$, we were unable to directly determine if there were seasonal variations in inorganic precursor trace gas emissions, however, AMS measurements of aerosol acidity may be used to infer these potential changes. During both IOPs,

anions and cations show good correlation (Figure S2), but in summer ammonium is insufficient for full neutralization of the anions, suggesting the aerosols in summer were more acidic. In most circumstances, ammonium nitrate will not partition into the condensed phase until particulate sulfate is fully neutralized (Guo et al., 2017). Thus the more acidic aerosol might be another explanation for the lower nitrate concentration in summer. In the spring IOP, ammonium is 13% higher than that required to fully balance

AMS-measured anions. This may be due to the presence of seasalt particles being transported to the SGP site in the spring; the AMS is not optimized for detection of seasalt and anionic species would be relatively more easily detected than Na and Ca in the particles. Alternatively, this difference may be due to slight measurement and calibration errors.

   The concentrations of biogenic VOCs are influenced by ambient temperature and sunlight, with higher

temperatures and more abundant sunlight, among other factors, producing higher emissions (Guenther et al., 2012). The average daily temperature at SGP was 24.0 °C during the summer IOP, which is significantly higher than during the spring IOP (15.9 °C). Days are also longer in the summer than in the spring. Concentrations of isoprene (m/z 69), a well-known biogenic VOC precursor of SOA, are about 2 times higher during the summer IOP compared to the spring IOP (Figure 3). As discussed in the

introduction, the design of two IOPs took into consideration the potential impacts of different stages and distribution of 'greenness' for cultivated crops, pasture, herbaceous, and forest vegetation types. Our isoprene observations suggest a complex relationship between emissions and vegetation. Isoprene concentrations scaled with temperature and sunlight, despite the fact that summer was significantly drier and that back trajectories suggest a smaller impact from biogenic-rich regions during summer IOP.

Monoterpenes (m/z 137), another category of biogenic VOCs emitted into the atmosphere, also show a similar pattern with higher concentrations observed in summer IOP. Back trajectories suggest more prevalent transport from the south in summer, which suggests higher anthropogenic impact. However, several representative anthropogenic VOCs observed by PTR-MS, including benzene, toluene and TMB, did not show significant enhancements during the summer season (Table 2). High concentrations of

benzene and toluene were observed during the first several days of summer IOP, but they were also accompanied by high levels of biogenic precursors (i.e., isoprene, monoterpenes) (Figure 3). During these time periods, back trajectories were from the southeast and the paths over the three-day period were generally short (Figure S3). Thus, the high concentrations were probably locally accumulated due to lower wind speeds. Acetonitrile, a key tracer for biomass burning, did not show significant changes during the

two IOPs. Therefore, the higher OA concentrations observed in the summer IOP relative to the spring are



likely related to more-intense biogenic emissions, rather than enhanced transport from urban areas or from biomass burning.

Recently, there has been intense research into the formation of highly-oxygenated organic compounds from biogenic precursors. This new group of highly-oxygenated molecule (HOMs) has been proposed to be the source of a major fraction of tropospheric submicron SOA (Bianchi et al., 2019;Ehn et al., 2017;Ehn et al., 2014). During the HI-SCALE campaign we observe indication that the bulk submicron OA contained a significant HOM fraction. Shown in Figure 4, the O:C ratios of submicron OA were mostly distributed in the range of 0.5-1.3 during the spring IOP, with an average of 0.84±0.14. The average of H:C ratios is 1.39, and the mean carbon oxidation state (OSc =2×O:C−H/C) is 0.29. The OA observed in summer IOP was significantly less oxygenated, with O:C, H:C and mean OSc values of 0.59±0.09, 1.52±0.11, and -0.34, respectively. There are several possible reasons for this relatively large difference in the O:C, H:C, and OSc values observed in the spring and summer season. The first possible explanation is that the aerosol in the spring is more aged due to a longer residence time in the atmosphere, potentially different oxidant concentrations, or a combination of both. Since photochemical aging leads to an increase in $f_{44}$ (Alfarra et al., 2004; de Gouw et al., 2005; Aiken et al., 2008; Kleinman et al., 2008), the level of $f_{44}$ can be considered as an indicator of atmospheric aging. Shown in the triangle plot (Figure 5A), the $f_{44}$ values in spring are generally higher than those in summer IOP, suggesting more aged aerosols arriving at the site in spring. A second possibility is that the VOCs contributing to SOA formation in summer are different than in spring. For example, we previously discuss that higher concentrations of isoprene and monoterpenes are observed in the summer, likely due to higher emissions, whereas concentrations of anthropogenics were more constant. Finally, it is possible that the more abundant biogenic VOCs in summer IOP were not transformed into a higher-oxygenated form in the aerosol phase, either due to differences in $RO_2$ radical chemistry, oxidants, or their residence time in the atmosphere (e.g., D'Ambro et al., 2017; Liu et al., 2016; Pye et al., 2019). Further identification of OA sources will be discussed in the next section via PMF analyses.

**3.3 Contributions of factors to organic aerosols in the spring- and summer- IOPs**

In order to better understand the sources of organic aerosol, we performed PMF analysis on the high resolution mass spectra data separately for each IOP. For the spring IOP, we chose a five factor solution (Figure S4). Shown in Figure S4, a first factor is characterized by an enhanced signal of $C_5H_6O^+$ (m/z 82), which is recognized as a tracer for IEPOX SOA (iSOA; Budisulistiorini et al., 2013). According to previous studies, a fractional $C_5H_6O^+$ signal ($f_{C5H6O}$) of 1.7 ‰ is roughly the background level (Hu et al., 2015), while the resolved first factor has an $f_{C5H6O}$ value of 4.5‰. This factor correlates with $SO_4$, with r = 0.55 (Figure 6). Based on the correlation of this factor with $SO_4$, and comparison of the mass spectrum to literature, we assign this factor as IEPOX-derived SOA. During the spring IOP, IEPOX-derived SOA factor contributed 27.9% of the total OA mass on average.



The second factor features a prominent marker at m/z 60 (primarily $C_2H_4O_2^+$) with a $f_{60}$ value of 0.9%. Based on previous studies (e.g., Cubison et al., 2011) this $f_{60}$ fraction is representative of air masses impacted by aged biomass burning. The trend of this factor (Figure 6B) tracks the time evolution of $C_2H_4O_2^+$ and $C_3H_5O_2^+$ well, with r values of 0.96 and 0.92, respectively. The signals of these two ions are thought to represent tracers for levoglucosan, which are also tracers for biomass burning organic aerosol

(BBOA) (Cubison et al., 2011; Jolleys et al., 2015). In addition, the time series of the second factor tracks a biomass burning event on April 29 well, with average $f_{60}$ value of 1.0% (details will be discussed in section 3.6). Thus we identify this second factor as BBOA. We also note that this BBOA factor has a relatively high $f_{44}$ value of 0.16, which further suggested the BBOA observed at SGP was aged. The BBOA factor accounts for 10.0% of total OA mass during the spring IOP on average, but at times can rise to over 50%

(Figure 6F).

We identify the third factor as HOA by comparison of the mass spectrum with literature spectra including the prominent signal at m/z 55 and 57 (Figure S4). This factor exhibits similar trend in time with toluene, a typical VOC tracer for primary emissions. Interestingly, the evolution of HOA doesn't correlate strongly with CO, a well known anthropogenic tracer. Shown in Figure S5, CO appears to be associated with both

HOA and BBOA, which likely impacts the correlation between CO and a single PMF factor. Our retrieval of an HOA factor from the PMF analysis contrasts with the results of a previous ACSM-based study at SGP. Based on 1.5-year of observational data, Parworth et al. (2015) suggested no HOA factor is extractable due to the rural characteristics of the SGP site. In this study, we retrieved an HOA factor, with an average contribution of 9.6% of total OA mass (Figure 6F). It is possible that the higher S/N and time

resolution of the HR-ToF-AMS used in our study relative to the quadrupole ACSM used in the Parworth study allows us to extract the HOA factor.

The fourth and fifth factors are two OOA (oxygenated organic aerosol) factors that are typically representative of SOA. The average mass spectra of the two OOA factors (Figure S4) show that the $f_{44}$ value is higher for OOA-2 (0.25) than OOA-1 (0.18), where the ion signal at m/z 44 commonly comes from

the thermal decomposition of carboxylic acids and other highly oxygenated compounds on the vaporizer. Therefore OOA-2 is identified as more-oxidized OOA (MO-OOA) and OOA-1 is identified as less-oxidized OOA (LO-OOA). However, we note that both factors have higher signal at m/z 44 than 43, indicating both are still signficantly oxidized. The mass spectra of LO-OOA factor reveal a prominent signal at m/z 91, the two dominant fragment ions at which include $C_7H_7^+$ and $C_3H_7O_3^+$. The $C_3H_7O_3^+$ ion

was proposed to be a tracer of isoprene photoixidation under low-$NO_x$ and low-RH conditions (Surratt et al., 2006), and $C_7H_7^+$ has been inferred to be a thermal decomposition product of dimers and oligomers in ISOPOOH-derived SOA (Riva et al., 2016). Shown in Figure 6D, the time series of both fragment ions track the evolution of LO-OOA well, suggesting the LO-OOA factor may indeed be associated with isoprene photooxidation chemistry. The MO-OOA factor correlates most strongly with acetone, which is an

oxidation product of several VOCs (Guenther et al., 2012). The sum of these two OOA factors contributed



approximately 50% of the total OA mass, indicating that the majority of the OA arriving at the site is relatively aged.

During the summer IOP, we chose a four-factor PMF solution, consisting of an IEPOX SOA factor, an HOA factor and two OOA factors (Figure 7). A BBOA factor was not identified during the summer season,

which is consistent with the low concentrations of BBOA observed in summer at SGP in a previous study (Parworth et al., 2015). The major contribution to the total OA mass is from the two OOA factors, the sum of which contributed >60% throughout the summer IOP. Similar to the spring IOP, OOA-1 is associated with enhanced signal at m/z 91 (Figure S6), and OOA-2 correlates best with acetone (Figure 7). However the $f_{44}$ values for both OOA factors are similar (0.11 vs. 0.12). The OSc values are somewhat different, -

0.264 vs.-0.099 for OOA-1 and OOA-2 respectively, though slightly smaller than the differences observed for the OOA-1 (-0.315) and OOA-2 (0.14) factors during spring. Considering the larger fraction of OOA-2 than OOA-1 in both spring (35.7% versus 13.7%) and summer (34.9% versus 28.4%), the smaller OSc values of OOA-2 factor in summer indicate that the OOA is in general less oxidized during the summer IOP. The contribution from HOA factor is minimal during the first several days of summer IOP (Figure 7)

but begins to increase after August 31 with the maximum approaching 50% during some periods. The transition in the HOA fraction is probably related to the transition of airmass origin from the northern region to the anthropogenic-rich regions to the south where primary emissions are stronger. The average contribution of the HOA factor during the summer IOP is 13.0%, higher than the fraction in the spring IOP. The higher summer HOA contribution is consistent with higher fraction of air masses originating from the

urbanized southern regions based on HYSPLIT analyses.

The fourth factor, IEPOX SOA, contributed 25.3% of the total OA mass, similar to spring IOP. Interestingly, the IEPOX SOA factor has similar $f_{44}$ values in spring and summer (0.116 vs. 0.137), but significantly different oxidation state (O:C ratios 1.349 vs. 0.653, OSc values 1.606 vs. -0.096). Possible reasons for this difference will be discussed in section 3.5, case studies on IEPOX SOA events.

**3.4 Comparison with previous studies at SGP**

Previous analyses have provided insights into the characteristics of aerosols impacting the SGP site and also showed that air mass origin has a large influence on aerosol properties. Using simultaneous trace gas measurements and back trajectory analyses in May of 2003, Wang et al. (2006) observed aged aerosols from wildfires at the SGP surface site, consistent with our results indicating biomass burning contribution

in the spring. Wang et al. (2006) also suggested high aerosol concentrations were strongly correlated with high $SO_2$ concentrations when the wind was from the east, south, and southeast, where several power plants are located. Although the $SO_2$ dataset was incomplete in this study, we did observe co-elevated OA and $SO_4$ concentrations in air masses from/traveling through the south region (details in section 3.5).

Based on 19-months of continuous measurements with a quadrupole Aerosol Chemical Speciation Monitor

(ACSM), Parworth et al. (2015) provided a more comprehensive description on the chemical composition



of submicron aerosols at the SGP site. Their seven-day back trajectory analyses showed that aerosol was transported to SGP during the spring from both the south and north, while southerly winds dominated during the summer. This seasonal variation is consistent with the higher fraction of southernly winds we observed during the summer IOP. The seasonal relative fractions of major AMS species, OA, SO$_4$, NO$_3$, and NH$_4$, are also similar between our study and theirs. However, Parworth et al. (2015) observed significantly lower absolute aerosol mass concentrations in the summer compared with the spring, which they attributed to temperature-dependent partitioning of semi-volatiles. In our study, only nitrate exhibited decreased concentrations from spring to summer, whereas both OA and SO$_4$ showed a ~50% increase. Another difference between the studies relates to the "triangle plots" for OA. Shown in Figure 5, we observe a sorting of the data according to season, with most $f_{44}$ values in the 0.10-0.30 range during spring and $f_{44}$ values lower than 0.20 observed during the summer. However in the Parworth study the OA was mostly "aged", across the entire observation period with most $f_{44}$ values concentrated in the 0.15-0.30 range. In summary, the Parworth study suggested the organic aerosols at the SGP site were in a more oxygenated state throughout the year, whereas the OA observed in our study demonstrated more seasonal variation.

There are also significant differences in the factor analysis results between the two studies. In the Parworth study, only three factors were isolated, including a BBOA factor and two OOA factors while we isolate an additional IEPOX factor in both seasons and an HOA factor in the spring. In the Parworth study, the OOA factor with a smaller $f_{44}$ value (OOA-2) was a larger fraction of the total mass in summer relative to spring, which is consistent with our observations that the summer OA appeared to be more fresh. It should be noted that the three factors retrieved in the Parworth study were based on data throughout the 19 months, and the contribution by BBOA was only evident in winter and spring seasons, which is again consistent with our results. One factor the Parworth study did not extract is the HOA factor, which they attributed to the rural setting of the SGP site. In our study, although the HOA only contributed ~10% of the total OA mass on average, the occasional spikes were accompanied with high concentrations of anthrogeponic tracers and is likely associated with transition of air mass origins. With respect to the IEPOX SOA factor, the Parworth study did take spatial distribution of isoprene emissions into consideration and suggested biogenic emissions likely contribute to SOA mass at SGP primarily during the summer, but this contribution was not directly attributed to IEPOX chemistry. However, Parsworth were not able to isolate an IEPOX SOA factor in their analysis. In our analyses, although the retrieved IEPOX SOA had similar fractions in spring and summer, the absolute mass concentration was indeed higher in summer, which agrees with the more intense biogenic emissions and photochemistry associated with higher solar insolation and temperatures as suggested in Parworth et al. (2015).

### 3.5 Case study 1: IEPOX SOA events

An IEPOX SOA factor was resolved during the spring and summer IOPs and it was a substantial contribution to the total OA mass. Here we selected one period of high IEPOX SOA (iSOA) from each IOP to describe in detail. For the spring IOP, we chose the period from May 8 20:00 to May 1018:00 (UTC)



(noted as spring iSOA event), and the period from September 4 6:00 to September 5 21:00 (UTC) for the summer IOP (noted as summer iSOA event).

The $C_5H_6O^+$ ion has been recognized as a unique marker for IEPOX SOA (Robinson et al., 2011; Lin et al.,
2012, 2013; Hu et al., 2015; Shilling et al., 2018). During both iSOA events, the time series of $C_5H_6O^+$ ion track that of IEPOX SOA and $SO_4$ (Figure 8A). Researchers have also identified additional ions that are representative of IEPOX SOA such as the ion of $C_3H_7O_2^+$ (Budisulistiorini et al., 2016), and we observe good correlation between $C_3H_7O_2^+$ (m/z 75) and $C_5H_6O^+$ (m/z 82) for both spring IOP and summer IOP. We also investigated the relationship between iSOA and its gas-phase precursors (Figure 8B). We attribute
the PTR-MS signal at m/z 71 as the sum of methyl vinyl ketone (MVK), methacrolein (MACR), and isoprene hydroxyhydroperoxide (ISOPOOH), all first-generation oxidation products of isoprene, and use the sum of m/z 69 (isoprene) and m/z 71 as an indicator of gas-phase iSOA precursors. During both events, generally speaking, the sum of isoprene and its first-generation oxidation products did not track the evolution of IEPOX SOA well, but at times an enhancement in IEPOX SOA followed after a peak in the
sum of m/z 69 and 71. Given the good correlation between $SO_4$ and IEPOX SOA, we speculate that the variation of IEPOX SOA at the SGP site was mostly driven by variations in the concentration and the acidity of particles, as iSOA precursors appear to be abundant during most of the campaign (Lin et al., 2012). Back trajectories provide some insights into the transport pathways associated with the iSOA events. In both the spring and summer iSOA events, air masses passed over urban areas before reaching the SGP
site, though there was some variation in the detailed trajectories. As seen in Figure S7, back trajectories indicate air masses during the spring iSOA event were uniformly transported from Houston, the refineries, and the shipping channels in Texas and the Gulf of Mexico. During the summer IOP, air masses originate in Missouri and Arkansas and then pass near Oklahoma City and Dallas/Fort Worth region before arriving at SGP. The computed trajectories are entirely within the boundary layer, indicating that surface emissions
should influence the air masses. In both cases, air masses pass through regions where $SO_2$ emissions are abundant, likely resulting in formation of acidic aerosol, which drives production of IEPOX SOA.

While the back trajectories during the spring iSOA event were mostly from Texas and the Gulf of Mexico to the south, summer back trajectories indicate air masses originate to the east. Parworth et al. (2015) suggested higher isoprene emissions originate from regions east of the SGP, which is supported by our
observations of higher m/z 69+71 in the summer iSOA event compared to the spring event (Figure 8B). Higher concentrations of isoprene and isoprene oxidation products in summer were accompanied by higher abundance of the $C_5H_6O^+$ ion, with the summer average $f_{C5H6O}$=6.51‰, much higher than that in spring ($f_{C5H6O}$=3.44‰). To roughly estimate the relative age and oxidation level of the aerosol during iSOA events, we present "triangle plot" similar to those described in Hu et al. (2015) in Figure S8. The iSOA
events group into distinct regions of these plots with the spring iSOA more similar to the OH-aged aerosols observed in southeast US (Hu et al., 2015) and the summer iSOA more similar to fresher aerosol. Note that the $f_{C5H6O}$ and $f_{CO2}$ values shown in Figure S8 are for the total OA, not just the IEPOX SOA factor. For





iSOA, the mean carbon oxidation state, OSc, was estimated to be -0.05 for summer event, and 0.75 for the spring event, respectively. Considering the higher levels of isoprene and its first-generation oxidation

products observed during the summer iSOA events, the higher $f_{C5H6O}$ value might suggest that the IEPOX pathway was favored over isoprene oxidation pathways producing higher generation, more oxidized products, such as ISOP(OOH)$_2$ (D'Ambro et al., 2017;Liu et al., 2016). In contrast, lower biogenic VOC concentrations in spring might result in a relatively higher ratio of oxidant-to-VOC, leading to the formation of more highly oxygenated organic aerosols.

Compared to other sites where IEPOX SOA were extensively studied, the IEPOX chemistry at SGP site appeared to be unique, especially for the iSOA event observed during spring IOP. Hu et al. (2015) summarized the characteristics of IEPOX SOA factors retrieved from a set of ambient observations covering urban, downwind urban, and pristine regions, and suggested a range of 12-40 ‰ of $f_{C5H6O}$ values for ambient IEPOX SOA. In this study, the summer IEPOX SOA had an $f_{C5H6O}$ value of 12.1‰, at the

lower end of previous results, whereas the $f_{C5H6O}$ value of spring IEPOX SOA was only 4.55‰. During the specific period of spring iSOA event, the $f_{C5H6O}$ values of 3.44‰ for bulk OA was also significantly lower than the average value of 6.5‰ suggested for ambient OA strongly influenced by isoprene emission (Hu et al., 2015). In addition, during the spring iSOA event, $f_{CO2}$ values of bulk OA were significantly higher than previously-reported results of ambient OA strongly impacted by isoprene emission. Recent work has

suggested some major IEPOX SOA components, specifically methyltetrol sulfates, may undergo further OH oxidation accompanied by the formation of HSO$_4^-$ ion (Chen et al., 2020; Lam et al., 2019). Thus, one possibility is that this mechanism produces more oxygenated IEPOX SOA in the spring, contributing to the higher $f_{CO2}$. Considering the relatively weak local emission of isoprene in spring at SGP, the higher $f_{CO2}$ but lower $f_{C5H6O}$ values might suggest aging processes of the IEPOX SOA during long-range transport. Since

most IEPOX SOA studies were previously conducted in the summer season with intense isoprene emissions, the distinct observations at SGP site, especially those during spring IOP, provided a unique point of view on SOA chemistry in rural environment with weak biogenic emissions.

**3.6 Case study 2: Biomass burning event on April 29**

In our PMF analysis, a BBOA factor was extracted during the spring IOP but not during the summer IOP,

in broad agreement with a previous observation of more substantial contribution from fire events during spring (Parworth et al., 2015). In this study we selected a specific biomass burning event to examine the evolution of typical tracers, sources and characteristics.

On April 29, 2016, two spikes of organic aerosol concentration were observed, as shown in Figure 1. From 00:00-9:00 (UTC) on April 29, OA is by far the dominant component of the aerosol phase, with an average

fraction of 79.13%. Particulate ammonium and nitrate also increased during these events likely due to increased emissions of their precursors, such as NH$_3$ and NO$_x$ (Paulot et al., 2017; Souri et al., 2017). No increase in sulfate concentrations was observed.



This period was identified as a biomass burning event based on enhanced signal at m/z 60 in AMS spectra, primarily composed of the $C_2H_4O_2^+$ ion. $C_2H_4O_2^+$, a fragment of levoglucosan and other plant carbohydrate

breakdown products (mannosan, galactosan), has long been recognized as a tracer for biomass burning. As seen in Figure 9, we observe a positive correlation between $C_2H_4O_2^+$ and OA during the spring campaign with a slope of 0.3%. During the BBOA event, this slope increased to 1.0%. According to previous studies, an m/z 60 ($C_2H_4O_2^+$) fraction of 0.3% is an approximate background level, representing air masses without biomass burning influence. An elevated fraction of $C_2H_4O_2^+$ suggests a significant contribution from biomass burning to OA (Cubison et al., 2011). Indeed, the PMF-resolved BBOA fraction also showed an

increase from the spring IOP average of 10% to as high as 77% on April 29.

Active fire data shows relatively intense fire hotspots north of the SGP site, with two concentrated areas in Kansas and Canada. Back trajectory analysis suggests that these emissions need to travel at least 6 hours before arriving at the SGP site (Figure S9). According to previous studies, levoglucosan has an atmospheric

lifetime of approximately 2 days (Hennigan et al., 2010), suggesting that a significant fraction of the levoglucosan may have decayed before reaching SGP, particularly for BBOA originating in Canada. The van Krevelen analysis (Figure S10) shows that the bulk organic aerosols are highly oxygenated, with an average O:C ratio of 0.839, much higher than typical characteristics of primary BBOA (Canagaratna et al., 2015), again consistent with the relatively long-range transport that was needed to bring BBOA to the SGP

site. Although the SGP site was strongly influenced by biomass burning, the oxidation state of OA during the April 29 event did not show significant differences relative to other periods in spring IOP, with OSc value of 0.281 (April 29 event) versus 0.289 (spring IOP average).

Interestingly, the elevated levels of $C_2H_4O_2^+$ and BBOA were not accompanied by significantly elevated levels of acetonitrile (m/z 42 measured by PTR-MS), which has been traditionally been used as a gas-phase

marker of biomass burning emissions. The average acetonitrile concentration for the spring IOP was 0.11 ppbv while during the biomass burning period it was 0.09 ppbv (Table 1). Even during the spikes in BBOA concentrations, acetonitrile concentrations were observed to only increase moderately, up to 0.2 ppbv (Figure S11). The specific reason we do not see a concomitant increase in acetonitrile with the clear BBOA plume remains ambiguous. We have ruled out the possibility that dilution and processing of acetonitrile

reduced its concentration below the PTR-MS detection limit; BBOA concentrations remain high despite dilution, acetonitrile has an atmospheric lifetime of 1.4 years, and the acetonitrile detection limit was 0.06 ppbv during the spring IOP. A second possibility is that the biomass burning did not emit significant amounts of acetonitrile, though we are unsure why this would be the case.

## 4. Conclusions and Discussions

Observations of gas-phase VOCs and particle-phase chemical composition were taken at the SGP site during the HI-SCALE campaign in 2016, with two intensive operation periods covering the spring and summer season. Aerosol and trace gases were characterized using the AMS and PTR-MS as well as a suite



of supplementary instruments. During both IOPs, organic aerosol is the most abundant particulate component. The OA concentrations were significantly higher during summer IOP and PTR-MS observations showed biogenic VOCs, including isoprene and monoterpene, were 2-3 times higher in the summer relative to the spring. PMF analyses were used to investigate aerosol sources and revealed that OOA was the dominant factor for both the spring and summer IOP. An IEPOX SOA factor was retrieved for both IOPs and contributed, on average, the second-largest fraction of the OA. The retrieved IEPOX SOA factor had lower levels of $f_{C5H6O}$ and higher $f_{CO2}$ in the spring relative to the summer, suggesting the IEPOX SOA was more aged in the spring than in the summer. Biomass burning events were only observed during the spring IOP, when they episodically contribute a significant fraction of the OA. The biomass burning aerosol was also found to have a higher oxidation state by the time it arrived at the SGP site, relative to other BBOA aerosol reported in the literature, likely due to aging occurring during the long-range transport. An HOA factor was also observed, but was a small fraction of the total OA, suggesting a limited contribution of fresh anthropogenic emissions during HI-SCALE. These observations suggest that biogenic emissions play the dominant role in formation of organic aerosol at the SGP site during HI-SCALE.

The SGP site is located in a rural setting, and biogenic emissions appear to largely control the concentrations of OA during the HI-SCALE campaign. In recent years, a number of studies have focused on HOMs produced from biogenic precursors, and these components are expected to play a key role in new particle formation (Ehn et al., 2014;Jokinen et al., 2015;Qi et al., 2018). The high oxidation state of the OA observed at the SGP site during HI-SCALE suggests that these molecules could indeed be important SOA components at SGP. Nevertheless, due to lack of molecule-level information, we were not able to quantitively evaluate the contribution of HOM chemistry to the oxidation state of OA at SGP site. Since HOMs likely contribute to new particle formation (NPF), they will also impact subsequent aerosol growth, CCN populations, and the influence of aerosols on global climate. Actually during HI-SCALE campaign, NPF events were more frequently observed in spring than in summer (Fast et al., 2019), in agreement with the more-oxygenated feature of OA observed during the spring IOP in this study. Considering the potential climate impacts, the highly-oxygenated nature of aerosols at the SGP site is an interesting topic which should be investigated further. The mixture of anthropogenic, biogenic, and biomass burning sources of OA at the SGP site provides an opportunity to rigorously evaluate explicit and parameterized treatments of a range of SOA pathway mechanisms.



**Author contribution:** J. Fast, L. Alexander and J. Shilling designed the experiments. J. Liu, L. Alexander and R. Lindenmaier carried out the measurements. J. Liu analysed the data and prepared the manuscript with contributions from all co-authors.

**The authors declare that they have no conflict of interest.**

**Acknowledgement**

The HI-SCALE field campaign was supported by the Atmospheric Radiation Measurement (ARM) Climate Research Facility and the Environmental Molecular Science Laboratory (EMSL), both of which are U.S. Department of Energy (DOE) Office of Science User Facilities sponsored by the Office of Biological and Environmental Research. Data analysis and research was supported by the U.S. Department of Energy (DOE) Office of Science, Office of Biological and Environmental Research, Atmospheric Systems Research (ASR) program. Pacific Northwest National Laboratory is operated for DOE by Battelle Memorial Institute under contract DE-AC05-76RL01830.



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



**Table 1.** Statistics of aerosol chemical composition measured by the AMS at the SGP site during the two Hi-Scale IOPs. Measurement units are μg/m³ for all species.

| | Spring IOP | | | | Summer IOP | | | |
|---|---|---|---|---|---|---|---|---|
| | mean | median | 25th percentile | 75th percentile | mean | median | 25th percentile | 75th percentile |
| Organic | 2.466 | 1.923 | 1.192 | 3.483 | 3.821 | 3.098 | 2.177 | 5.314 |
| Sulfate | 0.790 | 0.504 | 0.337 | 1.073 | 1.290 | 1.162 | 0.635 | 1.690 |
| Nitrate | 0.244 | 0.122 | 0.062 | 0.291 | 0.085 | 0.071 | 0.047 | 0.105 |
| Ammonium | 0.444 | 0.312 | 0.193 | 0.605 | 0.469 | 0.450 | 0.255 | 0.609 |
| Chloride | 0.011 | 0.008 | 0.006 | 0.012 | 0.012 | 0.010 | 0.005 | 0.016 |




**Table 2.** A summary of gas-phase VOCs observed at the SGP site by PTR-MS during the two Hi-Scale IOPs. Measurement units are ppbv for all VOC species.

| | Spring IOP | | | | Summer IOP | | | |
|---|---|---|---|---|---|---|---|---|
| | mean | median | 25th percentile | 75th percentile | mean | median | 25th percentile | 75th percentile |
| Isoprene | 0.267 | 0.188 | 0.131 | 0.283 | 0.513 | 0.384 | 0.249 | 0.622 |
| Monoterpenes | 0.098 | 0.078 | 0.062 | 0.117 | 0.255 | 0.186 | 0.106 | 0.313 |
| Acetone | 1.464 | 1.299 | 1.000 | 1.808 | 2.207 | 1.978 | 1.500 | 2.846 |
| Acetonitrile | 0.112 | 0.103 | 0.075 | 0.137 | 0.137 | 0.135 | 0.109 | 0.162 |
| Benzene | 0.082 | 0.066 | 0.046 | 0.101 | 0.128 | 0.105 | 0.068 | 0.152 |
| Toluene | 0.115 | 0.096 | 0.069 | 0.138 | 0.132 | 0.112 | 0.077 | 0.162 |
| TriMethyl-Benzene | 0.120 | 0.106 | 0.088 | 0.136 | 0.089 | 0.079 | 0.061 | 0.109 |



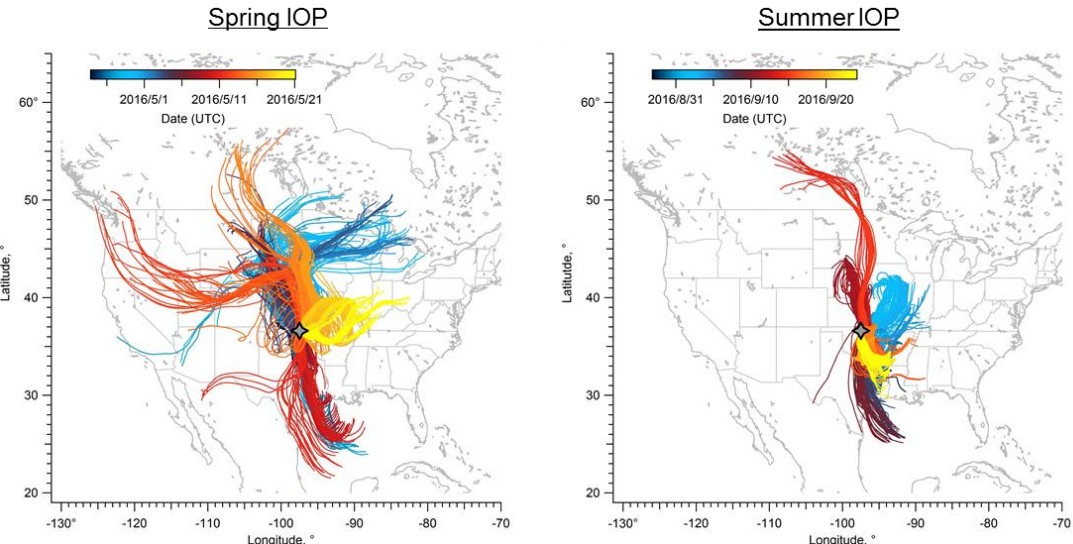

**Figure 1.** HYSPLIT back trajectories for spring and summer HI-SCALE IOPs. Each line shows a three-day back trajectory computed every three hours, with the SGP site (grey four-point star, ground level) as the end point of each trajectory. The lines are color-coded by date of arrival at the SGP site.

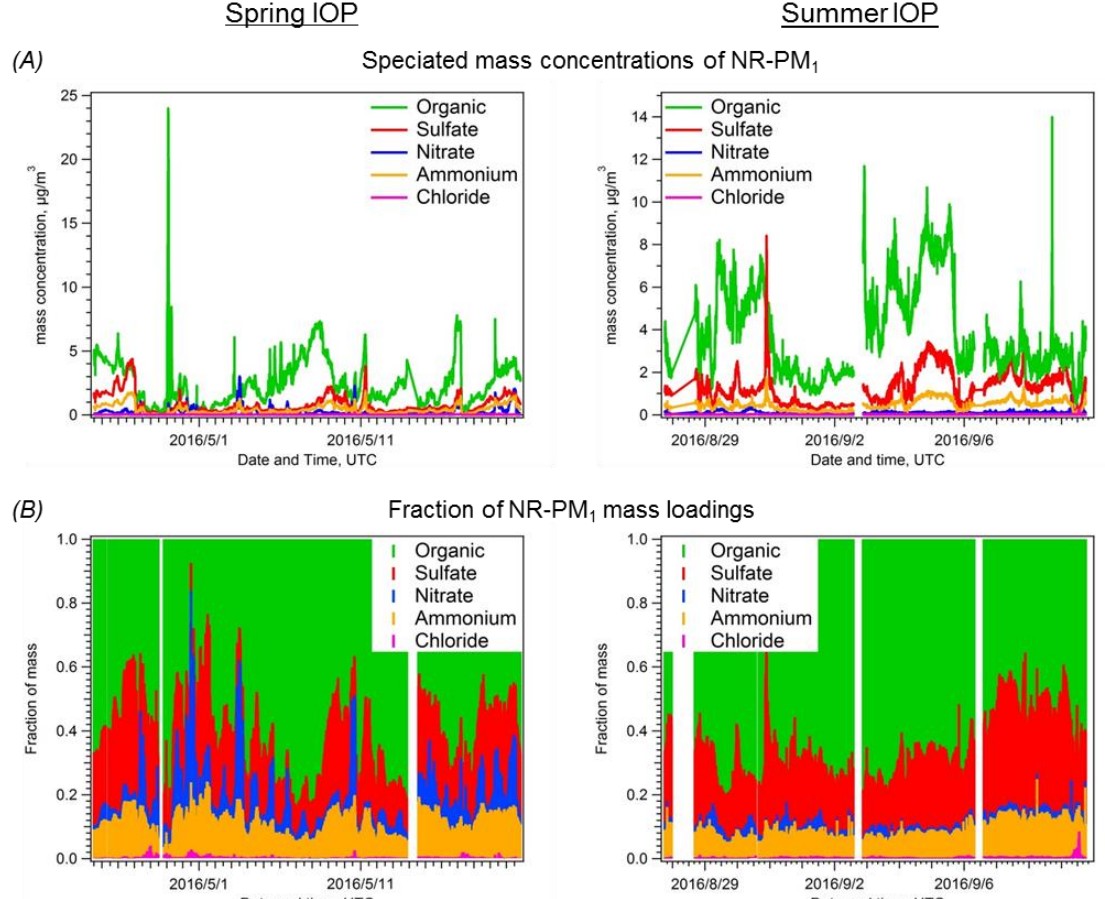

**Figure 2.** Time series of (A) absolute mass concentrations and (B) fractions of particle chemical composition, measured by the AMS for the spring- and summer- IOPs during the HI-SCALE campaign.



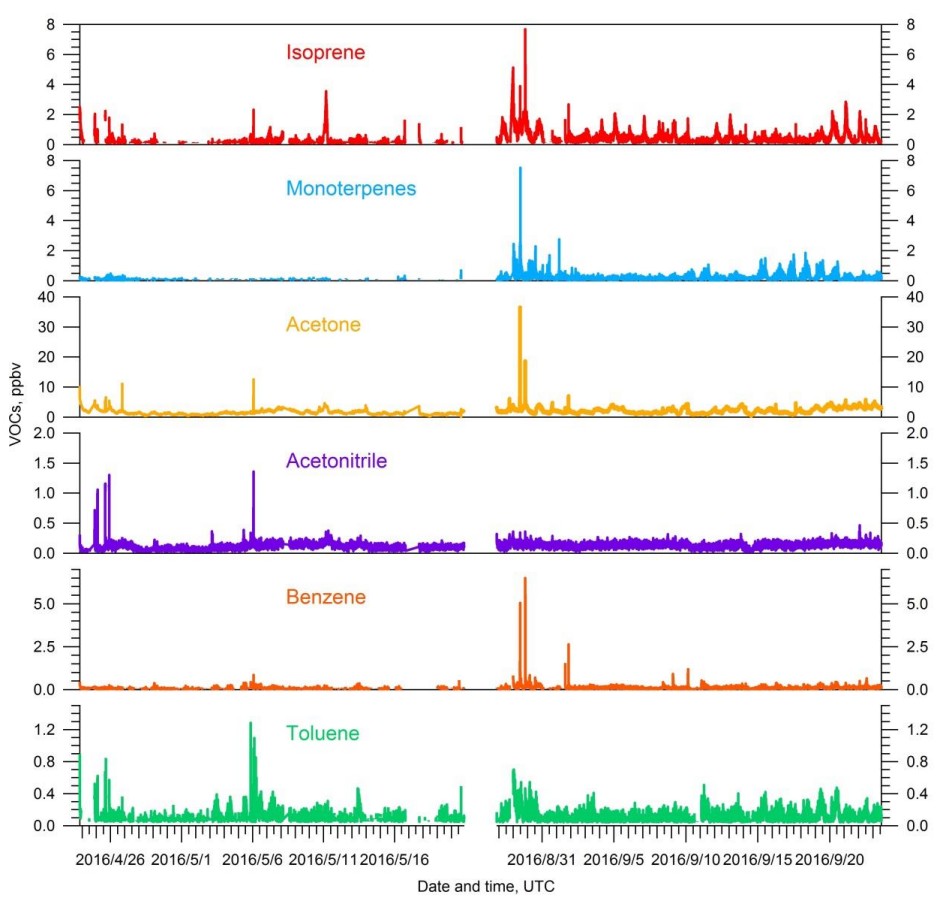

**Figure 3.** Time series of key VOCs, including isoprene, monoterpenes, acetone, acetonitrile, benzene and toluene, for both the spring- and summer- IOPs during the HI-SCALE campaign.






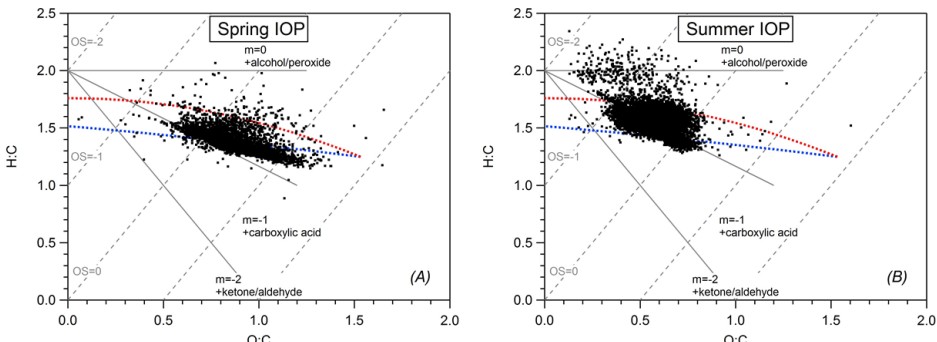

**Figure 4.** Van Krevelen diagrams of submicron OA observed at the SGP ground site during (A) the spring
IOP and (B) the summer IOP. Linear regression analyses showed a slope of -0.42 for spring IOP and -0.54
for summer IOP.



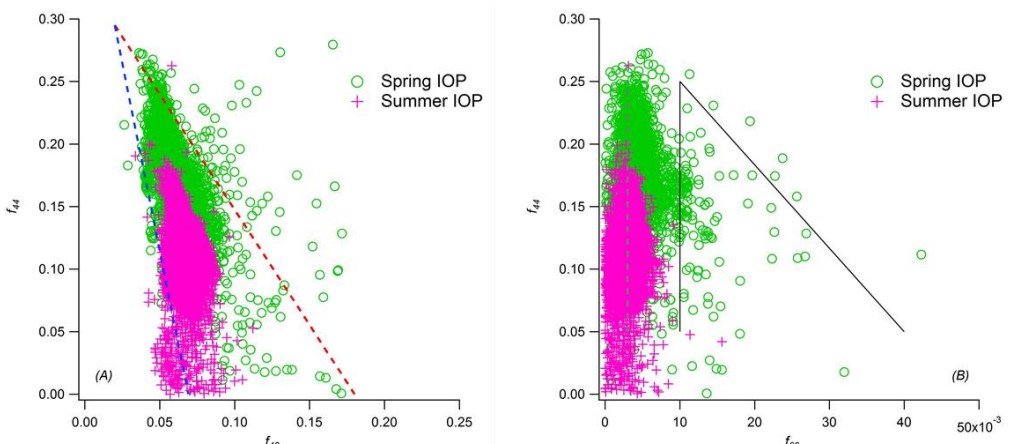

**Figure 5.** $f_{44}$ as a function of (A) $f_{43}$ and (B) $f_{60}$ for OA observed during spring- and summer- IOPs. $f_{44}$ represents the ratio of signal at m/z 44 (mainly $CO_2^+$) to the total organic signal, $f_{43}$ refers to ratio of total m/z 43 (mainly $C_3H_7^+$ and $C_2H_3O^+$) to the total organic signal, and $f_{60}$ refers to the ratio of m/z 60 (mainly $C_2H_4O_2^+$) to the total organic signal.



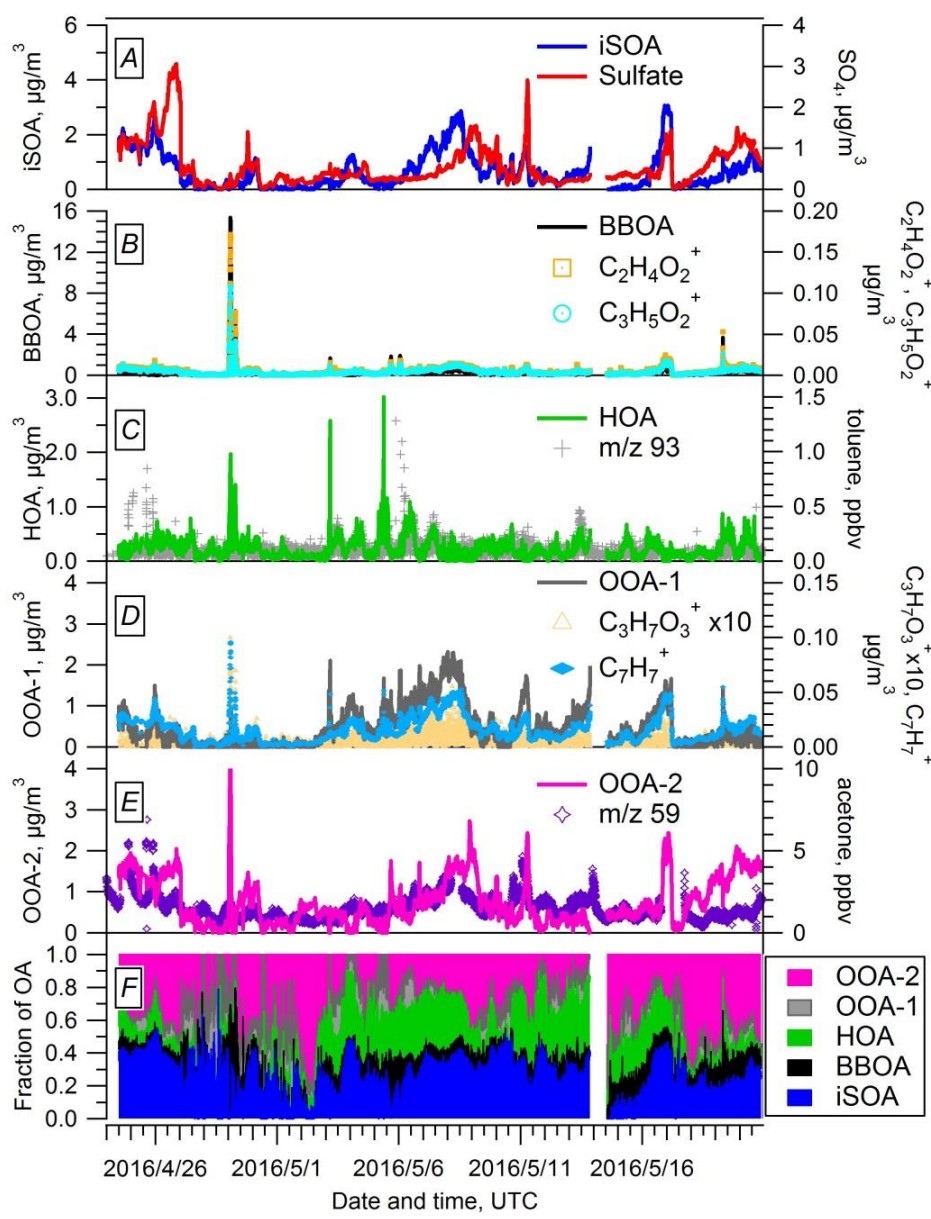


**Figure 6.** Time-series of PMF factors extracted from the AMS data for the spring IOP. Time-traces of additional species used to evaluate the identity of the PMF factors are also shown.

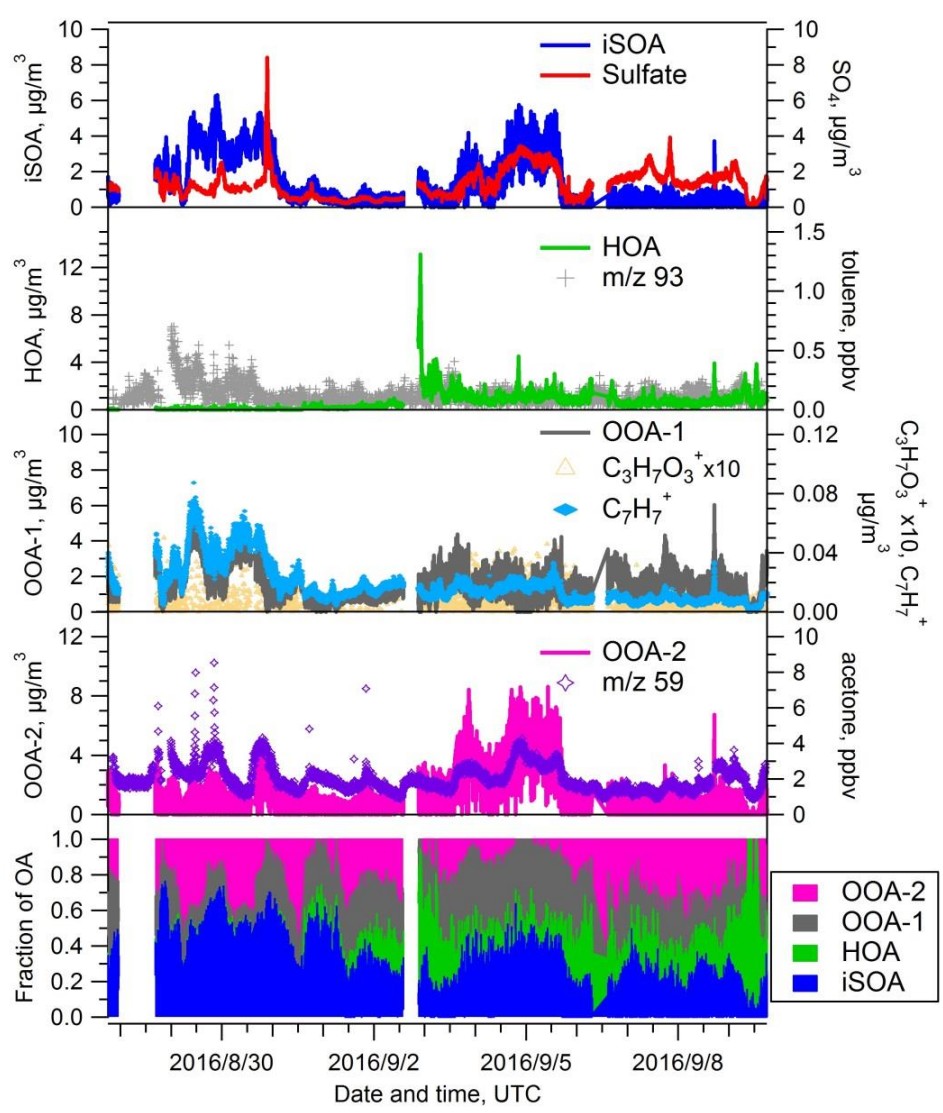

**Figure 7.** Time-series of PMF factors extracted from the AMS data for the summer IOP. Time-traces of additional species used to evaluate the identity of the PMF factors are also shown.

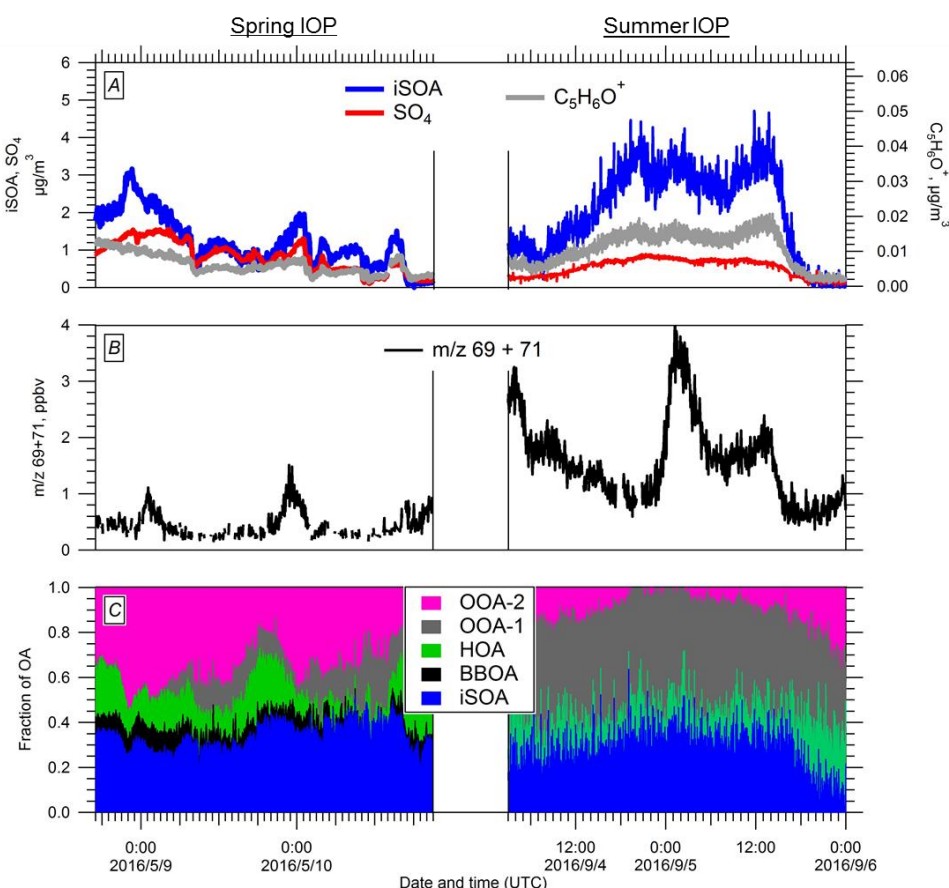

**Figure 8.** Temporal evolutions of (A) the iSOA PMF factor, $SO_4$ (left axis), $C_5H_6O^+$ (right axis), (B) the
sum of m/z 69 (isoprene) and 71 (first-generation oxidation products of isoprene), and (C) fractional
contribution of PMF-resolved factors to the total OA during the two iSOA events in spring- and summer-
IOPs, respectively.



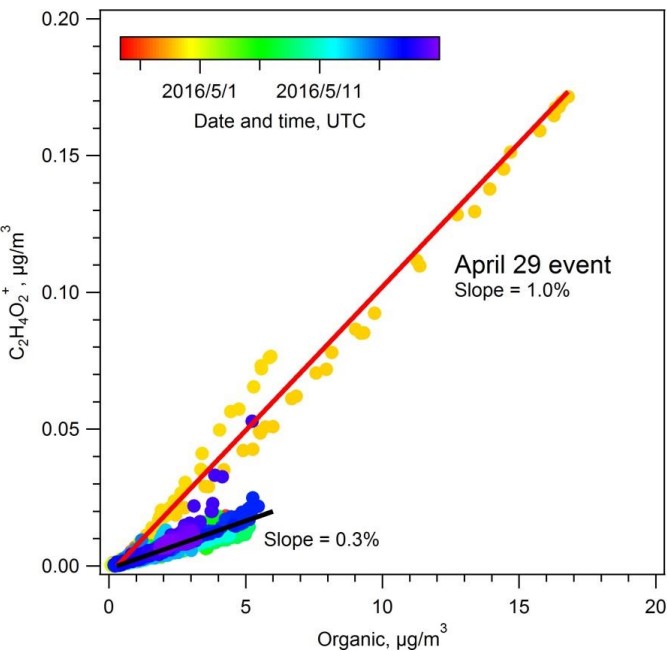

**Figure 9.** Scatter plot of the $C_2H_4O_2^+$ ion (a biomass burning marker) against total OA during the spring
IOP. The April 29 period is clearly differentiated from much of the data with a slope of 1.0%. The slope
for the remaining IOP data is 0.3%.