# Peer review of "Aerosol characteristics at the Southern Great Plains site during the HI-SCALE campaign"

_Atmospheric Chemistry and Physics, 2020_

## Referee Comment (RC1) · Anonymous Referee #1 · 2 Dec 2020

Review on "Aerosol characteristics at the Southern Great Plains site during the HI-SCALE campaign" by Liu et al. 2020

This research work presents the general aerosol characteristics at a rural site of north-central Oklahoma during HI-SCALE campaign. Generally, the results of this work was based aerosol mass spectrometer measurement. The chemical composition and OA PMF factor analysis during spring and summer intensive campaign were shown. The characterization of PMF factors: BBOA, isoprene-derived SOA, HOA and other OOAs were demonstrated and compared with previous studies.

This paper is well written and organized. I do not have many scientific and technical questions. The only thing is that I did not found very novel scientific finding in this paper. It is more like a report for the AMS measurement at north-central Oklahoma observation site, although the aerosol composition and source apportionment at this site has been reported previously as well. Some of the conclusions, which are based on deduction, are ambiguous. If this paper shall be published in ACP, I suggest it might be better to publish as a measurement report.

Other comments:

Line 177: It is hard to conclude the more acidic aerosol is another explanation for lower nitrate concentration since the acidity and ammonium nitrate partitioning influence each other. The author can give more accurate calculation on pH influenced on nitratre portioning as done in Guo et al. (Guo et al., 2016;Guo et al., 2017)

Line 179: I do not quite get this statement. The higher ammonium than that required to fully balance AMS-measured anions usually suggests potential presence of amine in aerosol phase (Docherty et al., 2011). And the anionic species in seasalt aerosols cannot be detected separately. Usually, they were detected as NaCl in AMS under high RH (Ovadnevaite et al., 2012).

Line 270: In addition to biogenic emission dominated areas, the m/z 91 was also found enhanced in the urban areas based on the spectrum of HOA (Ng et al., 2011) or from oxidation of aromatics. Since there were also strong anthropogenic emission influences in this site. The contribution from anthropogenic influences to m/z 91 is also one of the probabilities.

Line 398-403 What makes the IEPOX-SOA being through more oxidation process in spring than in summer? I just do not understand why the oxidation of methyltetrol tend to happen in spring compared to summer. Have the authors considered the impact of PMF analysis uncertainty to this factor analysis.

Line 443 Coggon et al. (Coggon et al., 2016) has reported that the emissions of nitrogen-containing VOCs (NVOCs) strongly depend on the fuel nitrogen content. They found markedly lower concentrations of acetonitrile for residential wood Burning. The authors can check if this is one of the reasons for low acetonitrile observed here.

Reference:

Coggon, M. M., et al.: Emissions of nitrogen-containing organic compounds from the burning of herbaceous and arboraceous biomass: Fuel composition dependence and the variability of commonly used nitrile tracers, Geophys Res Lett, 2016, 43, 9903-9912.

Docherty, K. S., et al.: The 2005 Study of Organic Aerosols at Riverside (SOAR-1): instrumental intercomparisons and fine particle composition, Atmos. Chem. Phys., 2011, 11, 12387-12420.

Guo, H., et al.: Fine particle pH and the partitioning of nitric acid during winter in the northeastern United States, Journal of Geophysical Research: Atmospheres, 2016, 121, 10,355-310,376.

Guo, H., et al.: Fine particle pH and gas–particle phase partitioning of inorganic species in Pasadena, California, during the 2010 CalNex campaign, Atmos. Chem. Phys., 2017, 17, 5703-5719.

Ng, N. L., et al.: Real-Time Methods for Estimating Organic Component Mass Concentrations from Aerosol Mass Spectrometer Data, Environ Sci Technol, 2011, 45, 910-916.

Ovadnevaite, J., et al.: On the effect of wind speed on submicron sea salt mass concentrations and source fluxes, Journal of Geophysical Research: Atmospheres, 2012, 117

---

## Referee Comment (RC2) · Anonymous Referee #2 · 7 Dec 2020

The paper by Liu et al. summarizes aerosol composition measurements during the HI-SCALE project in Oklahoma. Measurements of aerosol composition and trace gases were carried out using a high-resolution AMS and PTR-MS. HYSPLIT backtrajectories were used for source region characterization and positive matrix factorization was used to categorize OA into distinct factors. Since the campaign included a spring and summer segment, authors could contrast the seasonal composition differences. They conclude that overall biogenic VOCs seem to control OA formation. Case studies were explored in more detail to investigate OA formation from biomass burning and isoprene oxidation. Overall the paper is well written and fits the scope of ACP. Discussed seasonality in OA composition and formation pathways are interesting. There are some issues that need further clarification, especially in the assignment of PMF factors and

interpretation of those factors. Details are highlighted below.

1. There are many average values presented throughout the paper; it will be useful to also report the standard deviation of the averages to get a better understanding of the variability in the data. 2. L113: How was the AMS bounce fraction corrected for? Was SMPS data used solely for this instead of composition-based CE? If so, how were differences in upper size cut of the two instruments and density considered? 3. L180-181: what could the source of sea salt be at a region so far from the oceans? The majority of these particles will not make it through long-range transport? 4. L198: Another reason for low concentration of aromatics is their higher reactivity during the summer. This should be considered as a possibility. 5. Last paragraph in section 3.2: I find the discussion on oxidation state as measured by O/C, OSc and f44 circular. All of these metrics are different ways to show the same thing and primarily follow the f44 patterns, so the fact that O/C and OSc in spring were higher than in summer and then f44 in spring was also higher is not an additional evidence for increased oxidation state. 6. L245-250: BBOA factor: There's some enhancement in acetonitrile (as shown in SI) during the BB event in Spring. What is the correlation for the time series of BBOA and acetonitrile shown in Fig. S11. Furthermore, peaks in CO seem to correlate with peaks in BBOA factor; I wonder if correlation can be looked at for a subset of times (say when BBOA factor is higher than a certain amount or the time shown in S11) to have an external verification of the BBOA factor assignment. What was f73 in the BBOA factor? Was the fraction highest in the BBOA factor? If so, then it's circular to identify a factor based on f60 and f73 and show the great correlation of the factor with individual tracers at m/z 60 and 73. 7. Lines 251-261: the poor correlation between HOA and CO could also mean that CO has other sources. Given the influence of biogenic emissions, it's likely that secondary CO production is also contributing to the observed CO concentrations, which would certainly not be correlated with HOA. How is CO correlation with some of the BVOC tracers? 8. L259-261: I'm curious if the HR data of the HOA factor includes only the CxHy+ type of ions. Was the PMF run with the HR spectra or UMR? Given the use of HR-AMS, I assume the former although

there's no evidence of HR-type of PMF results. If HR spectra were not used in PMF, please explain why not. 9. L284-286: I thought OOA-1 is always the more oxidized type. The separation of OOA factors don't fully make sense to me. OOA-1 in Spring has a higher contribution of m/z 55, 57, etc, and lower f44, yet its OSc is higher than OOA-2. In summer, OOA-1 has a higher contribution of m/z 29, 43, 55, 57, similar f44 and lower OSc. It's confusing that in one season OOA-1 is more oxidized and in the other season it's OOA-2. Could it be that the two names are swapped (or one is a typo here)? Could it be that the OOA factors are not 'cleanly' separate from one another? 10. L429-430: If the lifetime of levoglucosan is 2 days, I don't think a significant amount of it would have decayed during a 6 hr transport time (remaining concentration=exp(-6/24) C0=0.88 of initial concentration). Please clarify. Perhaps you mean the transport time from Canada is longer, in which case that time should be noted here and not 6 hrs.

---

## Author Comment (AC1) · 2 Feb 2021

We thank the reviewer for the valuable comments and have addressed them carefully. Please see details in the attached files.

Please also note the supplement to this comment: https://acp.copernicus.org/preprints/acp-2020-1100/acp-2020-1100-AC1-supplement.zip

---

## Author Comment (AC2) · 2 Feb 2021

We thank the reviewer for the valuable comments and have addressed them accordingly. Please see attached files for details.

Please also note the supplement to this comment:
https://acp.copernicus.org/preprints/acp-2020-1100/acp-2020-1100-AC2-supplement.zip

---

## Author Response (AR1)

Dear editor,

This "author's response" file includes the point-by-point responses to the comments raised by the two reviewers, followed by main text and supplementary materials with changes highlighted. Here is a brief index:

Best regards,

Jiumeng

**Dear Editor,**

**Manuscript number:** acp-2020-1100

**Title:** Aerosol characteristics at the Southern Great Plains site during the HI-SCALE campaign

We thank the editor and reviewers for the valuable comments and suggestions. We respond to each of the reviewers' comments and suggestions below. We feel we have thoroughly addressed all comments. Reviewers' comments are in black, our response is in blue, and changes to the text follow in italics.

**Reviewer 1**

This research work presents the general aerosol characteristics at a rural site of north-central Oklahoma during HI-SCALE campaign. Generally, the results of this work was based aerosol mass spectrometer measurement. The chemical composition and OA PMF factor analysis during spring and summer intensive campaign were shown. The characterization of PMF factors: BBOA, isoprene-derived SOA, HOA and other OOAs were demonstrated and compared with previous studies.

This paper is well written and organized. I do not have many scientific and technical questions. The only thing is that I did not found very novel scientific finding in this paper. It is more like a report for the AMS measurement at north-central Oklahoma observation site, although the aerosol composition and source apportionment at this site has been reported previously as well. Some of the conclusions, which are based on deduction, are ambiguous. If this paper shall be published in ACP, I suggest it might be better to publish as a measurement report.

Our response: We thank the reviewer for the comments and have carefully considered publishing this as a measurement report. We feel that there is enough new science to justify publishing this as a regular manuscript. To our knowledge, there is only one other paper describing real-time aerosol chemical composition measurements at the ARM SGP site (Parworth et al 2015). The SGP site is the most heavily instrumented and longest-running of the three DOE-ARM supersites in the world and it represents an important rural continental environment. Understanding aerosol sources and properties at this type

of site is important for improving the representation of aerosols in earth system models. The main scientific findings of this paper are:

1) We evaluate sources of organic aerosol at the DOE ARM site and quantify HOA, OOA, BBOA, and IEPOX-SOA contributions to the total organic aerosol. We use backtrajectory analysis and real-time VOC measurements to understand the origin of these sources.

2) We report, for the first time, observations of VOC concentrations and their seasonal differences at the SGP site, which is important for modeling SOA formation.

3) We find that the OA composition observed at the SGP site was highly oxygenated, which helps to identify the chemical mechanisms contributing to SOA formation and aging at the site, such as the role of autooxidation generating HOMs. These observations will help improve representation of SOA at rural, continental sites.

In addition, we highlight the following differences from the Parworth 2015 paper, which is the only previous paper focusing on aerosol chemical composition at ARM's SGP site.

1) We observe different seasonal trends in aerosol composition than Parworth 2015.

2) The Parworth study did not have supporting VOC measurements, which characterizes the SOA source gases at the site and provides additional information on airmass origin.

3) We deployed an HR-ToF AMS at the site while Parworth deployed a quadrupole ACSM. The HR-ToF AMS allow us to better characterize the aerosol oxidation state and allowed us to identify additional PMF factors that were unresolved by Parworth.

4) OA in the Parworth study was somewhat more aged on average, whereas the use of HR-ToF AMS allowed us to distinguish the seasonal variation of OA oxidation states.

5) We separate and quantify the contribution of HOA and IEPOX-SOA PMF factors to the OA chemical composition at the SGP site. These important aerosol sources were not identified in the Parworth study.

We believe this paper worth a publication on ACP as a research article.

Other comments:

Line 177: It is hard to conclude the more acidic aerosol is another explanation for lower nitrate concentration since the acidity and ammonium nitrate partitioning influence each other. The author can give more accurate calculation on pH influenced on nitratre portioning as done in Guo et al. (Guo et al., 2016;Guo et al., 2017)

Our response: We are unsure of exactly what the reviewer is suggesting in this comment. The thermodynamics of sulfate/nitrate/ammonium mixtures are well understood and show that lower pH results in less nitrate partitioning into the condensed phase when other factors are held constant. We used ISORROPIA II to estimate the aerosol acidity and found that pH averaged 1.33±0.54 during the summer IOP and 2.28±0.78 during the spring IOP (the information has been added into the main text, please see Lines 178-180). The AMS-measured concentrations of anions and cations are used in these calculations and we do not have all measurements needed for fully understanding pH of aerosol (i.e., many gas phase measurements are missing). Nevertheless, the calculated pH supports our statement that aerosols are more acidic in summer. As shown in Figure R1, nitrate partitioning to the aerosol phase will be less favored at lower pH and aerosol was more acidic in summer than in spring.

[Figure]

*Figure R1. The "S curve", showing the relationship between $\varepsilon(NO_3^-)$ and pH calculated by ISORROPIA II for both spring and summer IOPs, in which $\varepsilon(NO_3^-)$ represents the particle-phase mass fractions of total nitrate (i.e., gas phase plus particle phase).*

Because of limitations in input data, the calculation was done in an "iteration" way. We use the measured aerosol-phase data as initial input, run ISORROPIA in the "forward" mode to predict gas-phase concentrations of $NH_3$, $HNO_3$ and HCl, and use the sum of predicted gas-phase and measured aerosol-phase concentrations as the input for next round. After ~ 20 rounds of iteration, the differences of predicted gas-phase concentrations from adjacent rounds and differences between predicted and measured aerosol-phase concentrations, were within 10%, i.e., comparable with measurement uncertainties. The results were further constrained with the $NH_3$ levels from nearby sites in the AMON network (Atmospheric Ammonia Network, http://nadp.slh.wisc.edu/amon/). The limited data input will of course bring in some uncertainties, but we believe the statement on more acidic aerosols in summer and the consequent influences on nitrate partitioning are valid.

Line 179: I do not quite get this statement. The higher ammonium than that required to fully balance AMS-measured anions usually suggests potential presence of amine in aerosol phase (Docherty et al., 2011). And the anionic species in seasalt aerosols cannot be detected separately. Usually, they were detected as NaCl in AMS under high RH (Ovadnevaite et al., 2012).

Our response: We agree that the presence of amines may be a possible explanation for our observations. We have added this into the discussion: "*This may be due to the potential presence of amines in the particle phase, as amines may contribute to fragments nominally assigned to $NH_4$ (Docherty et al., 2011)*" (Lines 186-187).

We also now recognize that our statement about seasalt aerosol was in error and have removed it from the text.

Line 270: In addition to biogenic emission dominated areas, the m/z 91 was also found enhanced in the urban areas based on the spectrum of HOA (Ng et al., 2011) or from oxidation of aromatics. Since there were also strong anthropogenic emission influences in this site. The contribution from anthropogenic influences to m/z 91 is also one of the probabilities.

Our response: We agree that in some literature studies, the m/z 91 was attributed to anthropogenic influence. In our PMF results $f_{91}$ was enhanced (~13‰) in the HOA spectrum. This is mostly attributed to the $C_7H_7^+$ fragment. As the reviewer stated, $C_7H_7^+$ can arise from biogenic SOA but can also arise from fragmentation of aromatic compounds; therefore, we introduced another fragment, $C_3H_7O_3^+$, a tracer of isoprene photoixidation. Both $C_7H_7^+$ and $C_3H_7O_3^+$ correlated with LO-OOA (Figure 6), supporting that the LO-OOA factor is associated with isoprene photooxidation chemistry. In addition, we found no correlation between m/z 91 and other anthropogenic tracers such as benzene and toluene. Thus for our discussion on LO-OOA here, we believe the influence of anthropogenic emissions to m/z 91 is minimal.

We have included it into our discussion (Lines 284-288).

Line 398-403 What makes the IEPOX-SOA being through more oxidation process in spring than in summer? I just do not understand why the oxidation of methyltetrol tend to happen in spring compared to summer. Have the authors considered the impact of PMF analysis uncertainty to this factor analysis.

Our response: Indeed, the PMF analysis might bring some uncertainties. However, the IEPOX factors in the spring and summer displayed significantly different mean carbon oxidation state in spring (1.61±0.003) than in summer (-0.10±0.005). These differences are large and unlikely to be due to irregularities in the PMF analysis. We note that we identify both an HOA and multiple OOA factors in both spring and summer, so it is relatively unlikely that the IEPOX-SOA factor is contaminated with unresolved HOA or OOA.

While we cannot definitively explain the reason for these differences, we explore several possible explanations in the manuscript. Seasonal changes in biogenic emissions show that the both isoprene and monoterpenes levels were lower in spring. This may allow some isoprene oxidation products to undergo further oxidation/aging process due to the relatively abundant oxidants during spring. Some recent studies, such as Chen et al. (2020), have proposed relevant mechanisms producing more oxygenated/functionalized

organosulfates from IEPOX products. Since most studies on IEPOX SOA were previously conducted during summer season, we believe this unique observation at SGP site during spring season provided a new perspective on SOA chemistry, especially in rural environment with limited emissions.

Line 443 Coggon et al. (Coggon et al., 2016) has reported that the emissions of nitrogen-containing VOCs (NVOCs) strongly depend on the fuel nitrogen content. They found markedly lower concentrations of acetonitrile for residential wood Burning. The authors can check if this is one of the reasons for low acetonitrile observed here.

Our response: We thank the reviewer for pointing this out and have included this as a possible reason for our observations.

We have adjusted the statements as: "*A second possibility is that the biomass burning did not emit significant amounts of acetonitrile. Acetonitrile emissions were reported to be significantly different among various biomass burning sources, with lower nitrogen containing biomass emitting less acetonitrile (Coggon et al., 2016). Thus it is possible that the fuel had lower nitrogen content.*" (Lines 468-471)

Our response: The AMS bounce fraction was corrected based on comparison with SMPS data. During HI-SCALE, the SMPS measured the particle size distribution from 14 nm to 710 nm (mobility diameter). The AMS CE drops to approximately 50% at 1 μm aerodynamic diameter, which corresponds roughly to a 700 nm mobility with a reasonable particle density of ~1.4g/cm$^3$. Therefore, the instruments are measuring very similar size ranges of particles. In addition, the comparison between SMPS-determined and AMS-determined aerosol volume concentrations (converted from mass concentrations of various species and composition-based density) showed a relatively constant ratio, as shown in the plot below, suggesting the relative abundance of various chemical compositions did not significantly influence the CE during our campaign.

[Figure]

*Figure R2. Aerosol volume concentrations determined from AMS measurements compared to measurements from the co-located SMPS.*

3. L180-181: what could the source of sea salt be at a region so far from the oceans? The majority of these particles will not make it through long-range transport?

Our response: We have eliminated these lines based on these concerns and the concerns of another reviewer.

4. L198: Another reason for low concentration of aromatics is their higher reactivity during the summer. This should be considered as a possibility.

Our response: We thank the reviewer for this comment. Yes, higher OH concentrations in the summer may also lead to lower concentrations of aromatics. We have added a line to the manuscript with this additional possibility (Lines 205-206).

5. Last paragraph in section 3.2: I find the discussion on oxidation state as measured by O/C, OSc and f44 circular. All of these metrics are different ways to show the same thing and primarily follow the f44 patterns, so the fact that O/C and OSc in spring were higher than in summer and then f44 in spring was also higher is not an additional evidence for increased oxidation state.

Our response: We thank the reviewer for raising this point, but disagree somewhat. OSc is the most complete measure of the carbon oxidation state and clearly it is influenced by O:C, which is in turn influenced by f44. But the H:C ratio also influences OSc and is independent of O:C. Additionally, high O:C isn't necessarily a function of a dominant f44; for example isoprene photooxidation SOA has high O:C yet m/z 44 is not the dominant peak in the spectrum (Liu et al., 2016). Therefore, we feel providing all these measures of the carbon oxidation state are useful for making the point, though perhaps the text is more wordy than necessary. To address the reviewer's concerns, we have shortened and simplified the discussion as follows (Lines 224-228).

*"The first possible explanation is that the aerosol in the spring is more aged due to a longer residence time in the atmosphere, potentially different oxidant concentrations, or a combination of both. Photochemical aging leads to an increase in f44, O:C, and OSc, all of which are higher in the spring than in the summer (Alfarra et 220 al., 2004; de Gouw et al., 2005; Aiken et al., 2008; Kleinman et al., 2008)."*

6. L245-250: BBOA factor: There's some enhancement in acetonitrile (as shown in SI) during the BB event in Spring. What is the correlation for the time series of BBOA and acetonitrile shown in Fig. S11. Furthermore, peaks in CO seem to correlate with peaks in BBOA factor; I wonder if correlation can be looked at for a subset of times (say when

BBOA factor is higher than a certain amount or the time shown in S11) to have an external verification of the BBOA factor assignment. What was f73 in the BBOA factor? Was the fraction highest in the BBOA factor? If so, then it's circular to identify a factor based on f60 and f73 and show the great correlation of the factor with individual tracers at m/z 60 and 73.

Our response: (1) Regarding the enhancement in acetonitrile in Fig. S11. There was some weak enhancement during the April 29 event, but this was not as clear of obvious as the increase in m/z 60 in AMS spectrum. The correlation between BBOA and acetonitrile is relatively weak, with $R^2 = 0.52$. Following the suggestions by another reviewer, we have added some discussions on the possibility of biomass burning types with lower acetonitrile emissions, such as residential wood burning (Lines 468-471).

(2) During the April 29 event period, CO did show a correlation with BBOA, but also with HOA (please see the plot below, Figure R3). It is reasonable that CO serves as a tracer for both biomass burning and anthropogenic emissions, so it is not surprising that periods influenced by biomass burning also display higher CO. We have added some of this text to the manuscript (Lines 256-257).

[Figure]

Figure R3. Time series of BBOA, HOA factors and CO concentration during the April 29 event.

(3) The *f73* we showed here is for the ion of $C_3H_5O_2^+$, which is also a widely used BB emission signature as this ion is related to levoglucosan-like species. The *f73* is highest in BBOA factor and averaged 6.3‰ in the BBOA factor.

7. Lines 251-261: the poor correlation between HOA and CO could also mean that CO has other sources. Given the influence of biogenic emissions, it's likely that secondary CO production is also contributing to the observed CO concentrations, which would certainly not be correlated with HOA. How is CO correlation with some of the BVOC tracers?

Our response: The poor correlation between HOA and CO could mean CO has other sources, and we have discussed that both BBOA and HOA could contribute to CO, resulting in the relatively weak correlation between CO and a single PMF factor (Lines 264-265 and Figure S5).

We examined the correlation between CO and BVOCs and do no find significant correlations (please see the two plots below, CO correlation with isoprene and monoterpenes, respectively). Therefore the secondary CO production is unlikely an important contributor to the observed CO concentrations.

[Figure]

*Figure R4. Comparison of CO with isoprene (left) and monoterpenes (right).*

8. L259-261: I'm curious if the HR data of the HOA factor includes only the CxHy+ type of ions. Was the PMF run with the HR spectra or UMR? Given the use of HR-AMS, I

assume the former although there's no evidence of HR-type of PMF results. If HR spectra were not used in PMF, please explain why not.

Our response: The PMF was run with the HR spectra. Shown in the plot below, the HOA factor is dominated by the $C_xH_y^+$ family of ions, but also has some contribution from the CHO family of ions. We compared the factor spectra with literature data, and found some previously reported hydrocarbon-like OA factors also include CHO type ions (e.g., Struckmeier et al., 2016; Hu et al., 2018), similar to our PMF results.

[Figure]

*Figure R5. HR spectra of PMF-resolved HOA factor during spring IOP.*

9. L284-286: I thought OOA-1 is always the more oxidized type. The separation of OOA factors don't fully make sense to me. OOA-1 in Spring has a higher contribution of m/z 55, 57, etc, and lower f44, yet its OSc is higher than OOA-2. In summer, OOA-1 has a higher contribution of m/z 29, 43, 55, 57, similar f44 and lower OSc. It's confusing that in one season OOA-1 is more oxidized and in the other season it's OOA-2. Could it be that the two names are swapped (or one is a typo here)? Could it be that the OOA factors are not 'cleanly' separate from one another?

Our response: Yes, this is a typo and has been corrected. We thank the reviewer for pointing this out.

10. L429-430: If the lifetime of levoglucosan is 2 days, I don't think a significant amount of it would have decayed during a 6 hr transport time (remaining concentration=exp(-6/24) C0=0.88 of initial concentration). Please clarify. Perhaps you mean the transport time from Canada is longer, in which case that time should be noted here and not 6 hrs.

Our response: Yes, we were referring to the transport time from the Canada fires. We have adjusted the expression accordingly (Line 446).

**Refereces**

Liu, J., D'Ambro, E. L., Lee, B. H., Lopez-Hilfiker, F. D., Zaveri, R. A., Rivera-Rios, J. C., Keutsch, F. N., Iyer, S., Kurten, T., Zhang, Z., Gold, A., Surratt, J. D., Shilling, J. E., and Thornton, J. A.: Efficient Isoprene Secondary Organic Aerosol Formation from a Non-IEPOX Pathway, Environ Sci Technol, 10.1021/acs.est.6b01872, 2016.

Hu, W., Day, D. A., Campuzano-Jost, P., Nault, B. A., Park, T., Lee, T., Croteau, P., Canagaratna, M. R., Jayne, J. T., Worsnop, D. R., and Jimenez, J. L.: Evaluation of the New Capture Vaporizer for Aerosol Mass Spectrometers (AMS): Elemental Composition and Source Apportionment of Organic Aerosols (OA), ACS Earth and Space Chemistry, 2, 410-421, 10.1021/acsearthspacechem.8b00002, 2018.

Struckmeier, C., Drewnick, F., Fachinger, F., Gobbi, G. P., and Borrmann, S.: Atmospheric aerosols in Rome, Italy: sources, dynamics and spatial variations during two seasons, Atmos. Chem. Phys., 16, 15277-15299, 10.5194/acp-16-15277-2016, 2016.

[revised manuscript text omitted]

**Aerosol acidity estimated using ISORROPIA II**

To estimate aerosol pH during spring and summer IOPs, ISORROPIA II was run with hourly-averaged data (including concentrations of aerosol-phase species, ambient temperature and relative humidity) as input. Hourly-averaged data were deployed considering that equilibrium states are typically achieved within 30 minutes under ambient conditions for submicron aerosols (Fountoukis et al., 2009). To simplify the simulations, ISORROPIA-II was run assuming particles are "metastable". It is also assumed that the particles are internally mixed and that pH does not vary with particle size (so that bulk properties represent the overall aerosol pH).

Because of limitations in input data, e.g., no gas phase $NH_3$ data available on site, and $SO_2$ only available for spring IOP, the calculation was done in an "iteration" way. We use the measured aerosol-phase data as initial input, run ISORROPIA in the "forward" mode to predict gas-phase concentrations of $NH_3$, $HNO_3$ and HCl, and use the sum of predicted gas-phase and measured aerosol-phase concentrations as the input for next round. After ~ 20 rounds of iteration, the differences of predicted gas-phase concentrations from adjacent rounds, and differences between predicted and measured aerosol-phase concentrations, were limited within 10%, i.e., comparable with measurement uncertainties. The results were further constrained with the $NH_3$ levels from nearby sites in the AMON network (Atmospheric Ammonia Network, http://nadp.slh.wisc.edu/amon/).

[Figure]

Figure S1. Map of the SGP site (green dot) and surrounding area. The plot is from the webpage of ARM SGP site (https://www.arm.gov/tour/sgp-overview.html).

[Figure]

Figure S2. Ion balance for both the spring- (left) and summer- (right) IOPs. Cation equivalence is calculated as $\frac{NH_4}{18}$, anion equivalence is calculated as $\frac{2 \cdot SO_4}{96} + \frac{NO_3}{62} + \frac{CHl}{35.5}$. The grey line indicates full neutralization.

[Figure]

Figure S3. 72-h HYSPLIT trajectory analyses of air arriving at the SGP site for the indicated days during the summer IOP. During these days, high concentrations of biogenic and anthropogenic VOC precursors were observed.

[Figure]

Figure S4. Mass spectral profiles of the 5-factor PMF solution chosen for the spring IOP data.

[Figure]

Figure S5. Time-series of BBOA, HOA and CO for the spring IOP.

[Figure]

**Figure S6.** Mass spectral profiles of the 4-factor PMF solution chosen for the summer IOP data.

[Figure]

Figure S7. 72-h HYSPLIT trajectory analyses for air arriving at the SGP site during the spring and summer IEPOX SOA events. The top panel shows the back trajectory for the days covering the spring iSOA event, while the bottom figures are for the summer iSOA event.

[Figure]

Figure S8. Scatter plot of $f_{CO2}$ and $f_{C5H6O}$ during the spring iSOA and summer iSOA events. The grey line represents background levels (quoted from Figure 5 in Hu et al., 2015).

[Figure]

Figure S9. Fire map retrieved from Terra/MODIS satellite observations for April 22-29, 2016 (left, created using © Google Earth), and NOAA HYSPLIT back trajectory paths for the biomass burning events observed at the SGP site on April 29, 2016 (right).

[Figure]

Figure S10. Van Krevelen plot of bulk organic aerosols for the spring IOP (black dots), and during the biomass burning event on April 29, 2016 (red circles).

[Figure]

**Figure S11.** Temporal evolution of AMS-reported chemical species, BBOA (resolved by PMF analyses), $C_2H_4O_2^+$, acetonitrile, and the mass fraction of all PMF-resolved factors during the April 29 biomass burning event.

**References**

Fountoukis, C., Nenes, A., Sullivan, A., Weber, R., Van Reken, T., Fischer, M., Matías, E., Moya, M., Farmer, D., and Cohen, R. C.: Thermodynamic characterization of Mexico City aerosol during MILAGRO 2006, Atmos. Chem. Phys., 9, 2141-2156, 10.5194/acp-9-2141-2009, 2009.